# MiR-221-3p/222-3p Cluster Expression in Human Adipose Tissue Is Related to Obesity and Type 2 Diabetes

**DOI:** 10.3390/ijms242417449

**Published:** 2023-12-13

**Authors:** Adriana-Mariel Gentile, Said Lhamyani, María Mengual-Mesa, Eduardo García-Fuentes, Francisco-Javier Bermúdez-Silva, Gemma Rojo-Martínez, Mercedes Clemente-Postigo, Alberto Rodriguez-Cañete, Gabriel Olveira, Rajaa El Bekay

**Affiliations:** 1Instituto de Investigación Biomédica de Málaga y Plataforma en Nanomedicina-IBIMA Plataforma BIONAND, 29580 Málaga, Spain; biogentile@gmail.com (A.-M.G.); saidlhamyani@gmail.com (S.L.); edugf1@gmail.com (E.G.-F.); javier.bermudez@ibima.eu (F.-J.B.-S.); gemma.rojo.m@gmail.com (G.R.-M.); gabolvfus@uma.es (G.O.); 2Clinical Unit of Endocrinology and Nutrition, University Regional Hospital of Málaga, 29009 Málaga, Spain; 3Spanish Biomedical Research Center in Physiopathology of Obesity and Nutrition (CIBERObn), Instituto de Salud Carlos III, 28029 Madrid, Spain; 4Andalucía Tech, Faculty of Health Sciences, Department of Systems and Automation Engineering, School of Industrial Engineering, Universidad de Málaga, Teatinos Campus, 29071 Málaga, Spain; maria.mengual.mesa@gmail.com; 5Unidad de Gestión Clínica de Aparato Digestivo, Hospital Universitario Virgen de la Victoria, 29010 Málaga, Spain; 6Centro de Investigación Biomédica en Red de Enfermedades Hepáticas y Digestivas (CIBEREHD), Instituto de Salud Carlos III, 28029 Málaga, Spain; 7The Spanish Biomedical Research Centre in Diabetes and Associated Metabolic Disorders (CIBERDEM), Instituto de Salud Carlos III, 28029 Madrid, Spain; 8Department of Endocrinology and Nutrition, Virgen de la Victoria University Hospital, 29010 Málaga, Spain; 9Institute of Biomedical Research in Málaga (IBIMA)-Bionand Platform, 29590 Málaga, Spain; 10Department of Cell Biology, Genetics, and Physiology, Faculty of Science, University of Málaga, 29010 Málaga, Spain; 11Unidad de Gestión Clínica de Cirugía General, Digestiva y Trasplantes, Hospital Regional Universitario de Málaga, 29010 Málaga, Spain; arodriguezcane@hotmail.com; 12Departamento de Medicina y Cirugía, Universidad de Málaga, 29010 Málaga, Spain; 13IBIMA-Plataforma Bionand, Hospital Regional Universitario de Málaga, 29010 Málaga, Spain

**Keywords:** miR-221-3p, miR-222-3p, miR-221-3p/222-3p cluster, human adipose tissue, obesity, type 2 diabetes

## Abstract

The progression of obesity and type 2 diabetes (T2D) is intricately linked with adipose tissue (AT) angiogenesis. Despite an established network of microRNAs (miRNAs) regulating AT function, the specific role of angiogenic miRNAs remains less understood. The miR-221/222 cluster has recently emerged as being associated with antiangiogenic activity. However, no studies have explored its role in human AT amidst the concurrent development of obesity and T2D. Therefore, this study aims to investigate the association between the miR-221-3p/222-3p cluster in human AT and its regulatory network with obesity and T2D. MiR-221-3p/222-3p and their target gene (TG) expression levels were quantified through qPCR in visceral (VAT) and subcutaneous (SAT) AT from patients (*n* = 33) categorized based on BMI as normoweight (NW) and obese (OB) and by glycemic status as normoglycemic (NG) and type 2 diabetic (T2D) subjects. In silico analyses of miR-221-3p/222-3p and their TGs were conducted to identify pertinent signaling pathways. The results of a multivariate analysis, considering the simultaneous expression of miR-221-3p and miR-222-3p as dependent variables, revealed statistically significant distinctions when accounting for variables such as tissue depot, obesity, sex, and T2D as independent factors. Furthermore, both miRNAs and their TGs exhibited differential expression patterns based on obesity severity, glycemic status, sex, and type of AT depot. Our in silico analysis indicated that miR-221-3p/222-3p cluster TGs predominantly participate in angiogenesis, WNT signaling, and apoptosis pathways. In conclusion, these findings underscore a promising avenue for future research, emphasizing the miR-221-3p/222-3p cluster and its associated regulatory networks as potential targets for addressing obesity and related metabolic disorders.

## 1. Introduction

Obesity has considerably increased in recent decades and is a huge public health problem in our society [1]. Obesity is generally accompanied by other metabolic diseases, such as type 2 diabetes (T2D), insulin resistance (IR), fatty liver, and cardiovascular diseases. Dysfunctional adipose tissue (AT) has been extensively proven to be involved in the development of these obesity-associated comorbidities [2,3].

AT is a highly dynamic endocrine tissue contributing to metabolic homeostasis [4]. The role of AT in energy storage is crucial for maintaining lipid homeostasis and avoiding ectopic fat accumulation [5]. Nonetheless, when it becomes severely dysfunctional and expands inappropriately to store excess energy, AT leads to metabolic alterations associated with obesity, such as IR and T2D [6,7]. This dysfunction is due to decreased vascularization (angiogenesis), increased hypoxia, and inflammation [8,9]. Thus, angiogenesis is considered a key process for healthy AT expansion under chronic overnutrition conditions, which prevents hypoxia and allows for the correct supply of nutrients to cells [10,11]. In this regard, it has been proposed that the onset of AT dysfunction is essentially determined by its capacity to store excess energy rather than by its size. Thus, there is inter-individual variability in AT expandability and remodeling that determines whether metabolically healthy status is maintained or not [12]. This would explain the existence of metabolically healthy obese subjects and lean subjects with metabolic disease [10,12,13,14,15]. 

MicroRNAs (miRNAs), small non-coding RNAs that regulate gene expression, have recently gained interest due to their role in the development of fat cells as well as in obesity and metabolic disorders [16]. Several studies have shown that the expression of miRNAs in preadipocytes (miR-221, miR-125b, miR-34a, miR-100, miR-130b, miR-210, and miR-185) is altered in obesity [17]. Also, our group has recently reported the dysregulation of several AT miRNAs in T2D and obesity, including miR-20b, miR-296, and Let-7f [18], and the involvement of other miRNAs (miR21) in AT functionality regulation [19]. 

MiR-221-3p and miR-222-3p are well described to be primarily involved in angiogenesis by controlling multiple angiogenic genes [12]. These miRNAs have been related to metabolic diseases in AT and as biomarkers [12,20,21,22]. In addition, miR-221 and miR-222 have been described to be novel diagnostic, prognostic, and therapeutic biomarkers in various diseases, including cancer and inflammatory diseases [23], and have been found to be upregulated in IR and certain types of cancer [24].

It has been also proposed that a complex miRNA regulatory network may control AT functionality, and that miRNA expression profile alteration could contribute to AT dysfunction related to T2D [24]. However, the mechanisms underlying the involvement of these miRNAs in AT regulation remain to be well determined. Taking into consideration the relevant role that the miR-221-3p/222-3p cluster can play in maintaining AT homeostasis, the present work aims to analyze the relationship between this cluster and its target gene network in human AT and obesity and T2D.

## 2. Results

### 2.1. Patient Characteristics

The anthropometric and clinical variables of the study subjects are summarized in Table 1. The subjects were divided into 3 groups consisting of normoglycemic normoweight (NG-NW), normoglycemic obese (NG-OB), and diabetic obese (D-OB) patients. To obtain homogeneous and reliable results, patients in each group were well matched in terms of age and sex. Significant differences were observed in the parameters used to differentiate the groups, such as BMI, blood glucose, homeostasis model of IR assessment index (HOMA-IR), triglycerides, and HDL-c. 

### 2.2. Human miR-221-3p and miR-222-3p Expression Profiles and Their Association with Obesity, T2D, and AT Depot

The study employed a general linear multivariate model with miR-221-3p and miR-222-3p expression levels as dependent variables, while tissue depot, degree of obesity, glycemic status, and sex were considered as independent variables. In order to reduce variability and improve the accuracy of the results, age and HDLc were introduced as covariates. This analysis demonstrated the discriminative power of these variables, as evidenced by the intercept of the model (Wilks’ λ = 0.848, F = 4.572, *p* < 0.015) and the significant effect of tissue depot (Wilks’ λ = 0.788, F = 6.869, *p* < 0.002) (Figure 1A).

Specifically, a significant intercept effect was observed (*p* = 0.011 for miR-221-3p and *p* = 0.005 for miR-222-3p) as well as a significant tissue depot effect (*p* = 0.039 for miR-221-3p and *p* = 0.001 for miR-222-3p) (Figure 1A). On the other hand, by means of pairwise comparisons, we were able to show the effect of obesity (OB vs. NW) (F = 5.990; *p* = 0.018 for miR-221-3p); the effect of diabetes (F = 5.661; *p* = 0.021); and the effect of tissue (VAT vs. SAT) (F = 6.801; *p* = 0.012 for miR-221-3p and F = 13.868; *p* = 0.0001 for miR-222-3p) (Figure 1B). To delve deeper into the impact of obesity, tissue type, and T2D on miRNA expression, separate linear regression models were applied. In these models, the expression of each miRNA was treated as a dependent variable, while obesity, tissue type, T2D, sex, and age served as independent variables.

The regression model for miR-221-3p (R2 = 0.200; F = 4.255; *p* = 0.002) indicated that 20% of the variation in miRNA-221 expression could be attributed to the positive effect of obesity (β = 0.268; *p* = 0.041) and the variance among different tissue types (β = −0.310; *p*= 0.007) (Figure 1A,B and Appendix A). Likewise, the regression model for miR-222-3p (R2 = 0.188; F = 4.003; *p* = 0.003) suggested that 18.8% of the variation in miRNA-222-3p expression could be explained by the variance among different tissue types (β = −0.442; *p* = 0.0001) (Figure 1A,B and Appendix A). The histogram and Q-Q plot showed that all of the parameters tested had normal distribution (Appendix A). These findings highlight that, in the presence of both obesity and T2D, the expression levels of miR-221-3p and miR-222-3p are contingent on the type of tissue (Figure 1A,B). 

### 2.3. Human miR-221-3p and miR-222-3p Expression Profiles Were Analyzed in Relation to Obesity, T2D, and AT Depot, with Consideration of Sex-Based Differences

Given the known variations in fat distribution and influence of sex hormones, it was hypothesized that the roles of these miRNAs in subcutaneous adipose tissue (SAT) and visceral adipose tissue (VAT) might differ by sex. Accordingly, the statistical analyses were stratified by sex. Figure 1C illustrates that the expression levels of miR-221-3p and miR-222-3p within each patient group, in the same tissue type, did not exhibit statistically significant differences by sex.

However, Figure 1D demonstrates a significant difference in the expression of miR-221-3p within the female NG-NW group when comparing VAT and SAT.

Additionally, Figure 1D reveals a significant variation in the expression of miR-222-3p among both female and male individuals in the NG-NW group. Furthermore, a significant difference was observed in the female D-OB group.

### 2.4. Mouse miR-221-3p and miR-222-3p Expression Profiles and Their Association with Obesity, T2D, and AT Depot

A general linear multivariate model, including the intercept, was employed to analyze miR-221-3p and miR-222-3p expression levels as dependent variables, with tissue depot, degree of obesity, and glycemic status as independent variables. This approach aimed to demonstrate the discriminative power of these variables. The model revealed significant differences in the intercept (Wilks’ λ = 0.373, F = 23.497, *p* < 0.0001), tissue depot (Wilks’ λ = 0.762, F = 4.382, *p* < 0.022), and diabetes (Wilks λ = 0.716, F = 5.549, *p* < 0.009), and an interaction effect between diabetes and tissue depot (Wilks’ λ = 0.757, F = 4.501, *p* < 0.020), as shown in Figure 2A.

Specifically, a significant intercept effect was observed (*p* = 0.0001 for miR-221-3p and *p* = 0.0001 for miR-222-3p); a significant tissue depot effect was observed (*p* = 0.007 for miR-221-3p); a significant diabetes effect was observed (*p* = 0.002 for miR-221-3p and *p* = 0.0001 for miR-222-3p); a significant interaction effect was observed between diabetes and tissue (*p* = 0.012 for miR-221-3p) (Figure 2A). On the other hand, by means of pairwise comparisons, we were able to show the effects of obesity (OB vs. NW) (*p* = 0.045 for miR-221-3p and *p* = 0.017 for miR-222-3p); the effect of diabetes (D vs. NG) (*p* = 0.0001 for miR-221-3p and *p* = 0.0001 for miR-222-3p); the effect of tissue (VAT vs. SAT) (*p* = 0.016 for miR-221-3p) (Figure 2B). 

To further dissect the impact of obesity, tissue type, and T2D on miRNA expression, linear regression models were applied. Each miRNA was treated as a dependent variable in separate models, with obesity, tissue type, and T2D as independent variables.

The regression model for miR-221-3p (R^2^ = 0.308; F = 6.038; *p* = 0.002) indicated that 30.8% of the variation in miRNA-221 expression could be attributed to the positive effect of diabetes (β = 0.514; *p* = 0.004) and the variance among different tissue types (β = −0.332; *p* = 0.027) (Figure 2 and Appendix A). In the case of miR-222-3p, the regression model (R2 = 0.274; F = 5.279; *p* = 0.005) suggested that 27.4% of the variation in miRNA-222-3p expression could be explained by the positive effect of diabetes (β = 0.441; *p* = 0.015) (Figure 2 and Appendix A). The histogram and Q-Q plot showed that all of the parameters tested had normal distribution (Appendix A).

These findings indicated that, in the presence of both obesity and T2D, the expression levels of miR-221-3p and miR-222-3p are contingent on the type of tissue (Figure 2).

To understand the potential impact of obesity and diabetes on a specific AT location, we analyzed miRNA expression based on tissue type. Each miRNA was treated as a dependent variable in separate models, with obesity and T2D as independent variables categorized by tissue type (Appendix A–D).

### 2.5. In Silico Analysis of miRNA-221-3p/222-3p Target Genes

The in silico analysis utilizing miRTarBase revealed a total of 911 validated target genes (TGs) for miR-221-3p and miR-222-3p. Among these, 53 TGs were found to be common for both miRNAs (Table 2 and Appendix A). Pathway analysis conducted with PANTHER indicated that miR-221-3p and miR-222-3p target genes predominantly participate in various cellular processes, including angiogenesis, apoptosis, inflammation, and adipogenesis, as well as the PDGF, interleukin, and Wnt signaling pathways (Figure 3A). Furthermore, analysis using miRWalk identified 17 predicted TGs for the mature sequences of both miR-221-3p and miR-222-3p that were specifically involved in angiogenesis, apoptosis, and adipogenesis (Table 2 and Appendix A).

The miRTarBase 4.0 database was used to predict validated TGs for miR-221-3p and miR-222-3p. The miRWalk 2.0 database was used to predict non-validated TGs involved in adipogenesis, angiogenesis and apoptosis signaling pathways. GeneCodis3 was used for enrichment analysis. A total of 24 putative candidates were highlighted and the expression levels of 3 TGs (*DVL2*, *ETS*, *IL1rap*) were analyzed in VAT and SAT by qPCR analysis.

### 2.6. Annotation Enrichment Analysis of miR-221-3p/222-3p Target Genes

Enrichment analysis was conducted using GeneCodis3, revealing a total of 24 target genes (7 validated, 17 predicted). Notably, 3 of these genes (disheveled segment polarity protein 2 (*DVL2*), E26 transformation-specific (*ETS*), and interleukin-1 receptor accessory protein (*IL1rap*) associated with angiogenesis and apoptosis signaling pathways exhibited the highest scores and specificity (Figure 3B). Figure 3C illustrates the potential interactions among these target genes within these pathways.

### 2.7. Identification of miRNA Binding Sites within Target Genes

While most miRNAs bind to mRNA through canonical sites, some utilize non-canonical sites, albeit with varying effectiveness [25]. TargetScan Human, a tool used in this study, allows for the detection of binding sites, and its site-recognition predictions have been reported to be comparable to in vivo approaches [25]. The context++ score percentile value quantifies the effectiveness of each binding site. Additionally, the probability of conserved targeting (PCT) is employed to gauge the biological relevance of predicted miRNA–target interactions [26]. PCT values range between 0 and 1, with higher values indicating greater conservation and a higher likelihood of detectable biological functions [26].

Our findings demonstrated that miR-221-3p exhibited effective binding sites within ETS1, DVL2, and IL1RAP (context++ score percentiles: 70%, 65%, and 94%, respectively). Similarly, miR-222-3p also displayed effective binding sites within the same genes (context++ score percentiles: 72%, 70%, and 94%, respectively). Furthermore, the PCT value, indicative of the conservation probability for a single target site [26], was measured at 0.10, suggesting evolutionary conservation (Appendix A).

### 2.8. Gene Expression of ETS1, DVL2, and IL1RAP in Human VAT and SAT: Impact of Obesity, Glycemic Status, and Sex

A general linear multivariate model, which included the intercept, was employed to assess the expression levels of *DVL2*, *ETS1*, and *IL1RAP* as dependent variables. Tissue depot, degree of obesity, sex, and glycemic status were treated as independent variables. In order to reduce variability and improve the accuracy of the results, age and HDLc were introduced as covariates, revealing significant discriminative power (Wilks’ λ = 0.839, F = 3.208, *p* < 0.025). Additionally, interaction effects were observed among tissue, obesity, and sex (Wilks’ λ = 0.831, F = 3.381, *p* = 0.025). Specifically, intercept effects were detected for *DVL2* (*p* = 0.017) and *ETS1* (*p* = 0.011); a tissue effect was detected for *IL1RAP*; an interaction effect was observed between tissue depot, obesity, and sex for *DVL2* (*p* = 0.019); an interaction effect was observed between obesity and sex for *ETS1* (*p* = 0.025); and an interaction effect was detected between tissue, diabetes, and sex for *DVL2* (*p* = 0.025) (Figure 4A).

To further analyze the impact of obesity, tissue type, sex, and glycemic status on the expression of target genes, linear regression models were utilized. Each target gene expression was taken as a dependent variable in separate models, while obesity, AT type, sex, and glycemic status served as independent variables. The regression model for *IL1RAP* (R^2^ = 0.155; *p* = 0.034) revealed that 15.5% of the variation in its expression could be attributed to the variance among different tissue types (β = −0.267; *p* = 0.027) (Figure 4A).

Furthermore, a Student’s *t*-test was applied to compare the expression levels of *DVL2*, *ETS1*, and *IL1RAP* in VAT and SAT among male and female individuals (Figure 4B). The results demonstrated significant differences in *DVL2* and *ETS1* expression (*p* = 0.039 and *p* = 0.037, respectively) between male and female patients in SAT of the NG-OB group (Figure 4B). Additionally, *IL1RAP* expression displayed a statistically significant difference (*p* = 0.018) in SAT of DOB patients (Figure 4B).

Taken together, these findings indicated that in the presence of sex, obesity, and T2D, the expression levels of *DVL2*, *ETS1*, and *IL1RAP* are dependent on the type of AT depot (Figure 4A,B).

## 3. Discussion

Our results showed that the expression levels of miR-221-3p and miR-222-3p are tissue-dependent and related to both obesity and T2D. Consistent with our findings, miR-221-3p has been shown to be involved in promoting AT inflammation [27,28]. Moreover, both miR-221-3p and miR-222-3p have been implicated in contributing to cardiovascular pathology by influencing fat and glucose metabolism [29] and are associated with various aspects, such as fat depots, obesity, IR, and T2D [20]. Although our data aligned with these previous findings, our study provides novel evidence that miR-221-3p/222-3p cluster expression in human AT could be related to obesity and fat depots when T2D is considered. To the best of our knowledge, this is the first study describing a possible connection between obesity, T2D, and miR-221-3p/222-3p cluster expression in human VAT and SAT in both sexes, male and female. This relationship could be mediated through some signaling pathways involved in AT functionality regulation. 

As is widely recognized, a single miRNA can modulate the expression of multiple genes, either amplifying or attenuating their expression through canonical or non-canonical pathways [30]. Furthermore, a specific gene can be under the regulation of several miRNAs. This complexity can result in gene and miRNA expression patterns that are not identical. MiRNAs and their regulated genes may display different temporal responses [30]. MiRNA expression can undergo rapid changes in response to cellular signals, whereas gene expression may be slower and more prolonged. Consequently, miRNAs may demonstrate expression patterns that are opposite to the genes they regulate [30]. Thus, our results show the patterns that the miR-221-3p/miR-22-3p cluster and its respective target genes and pathways can exhibit.

Our study underscores that miR-221-3p and miR-222-3p expression levels are contingent on tissue type and associated with both obesity and T2D. These findings align with previous research indicating the involvement of miR-221-3p in promoting AT inflammation and its collaboration with miR-222-3p in impacting cardiovascular health through their effects on fat and glucose metabolism [29]. Additionally, they have been linked to various aspects of AT regulation, including fat distribution, insulin resistance, and T2D [20]. Building upon these insights, our study introduces novel evidence suggesting a potential connection between obesity, T2D, and the expression of the miR-221-3p/222-3p cluster in human VAT and SAT across both male and female subjects. 

Furthermore, our bioinformatics analysis identified 24 target genes shared by both miR-221-3p and miR-222-3p. Among these, E26 transformation specific-1 (ETS1), disheveled 2 (DVL2), and interleukin 1 receptor accessory protein (IL1RAP) play critical roles in regulating AT functionality through pathways involved in angiogenesis, apoptosis, and Wnt signaling. DVL is a cytoplasmic adaptor protein crucial for Wnt signaling, influencing cell proliferation and fate decisions, while also playing a role in noncanonical Wnt signaling governing cell polarity and migration [31]. ETS1, a member of the ETS transcriptional factor family, participates in cellular differentiation, tissue remodeling, angiogenesis, and tumorigenesis. It is recognized both as an oncogene and for its apoptosis-promoting activity [32]. The IL-1 family is a group of cytokines that play a central role in the regulation of immune and inflammatory responses [33]. Interleukin-1 is a key inflammatory cytokine that mediates its effects through a type I receptor and receptor accessory protein. These two molecules are members of a wider family of proteins that have in common the presence of immunoglobulin domains in the extracellular region of the protein and a TIR domain in the cytoplasmic region [34].

Moreover, these target genes exhibited varying expression patterns based on obesity, AT depot, and sex, when glycemic status was considered. The miR-221-3p/222-3p cluster has demonstrated involvement in apoptosis, Wnt signaling, and angiogenesis through the regulation of several target genes [35,36,37,38]. Specifically, *ETS1* and *DVL2* are known regulators of the WNT signaling pathway and angiogenesis [39,40], while *IL1RAP* is implicated in apoptosis regulation [41]. Notably, we conducted a multivariate analysis considering obesity, diabetes, sex, and AT depot, shedding light on the complex interactions between miR-221 and miR-222 within tissues. This emphasizes the importance of studying the miR-221-3p/222-3p cluster within the context of these interdependent factors. 

Furthermore, our study introduces the finding that in human AT, *DVL2* expression is influenced by both fat depot, diabetes, sex, and obesity, aligning with previous research linking this gene to insulin sensitivity. We also provide new evidence suggesting that *ETS1* and *IL1RAP* expression levels may be related to fat depots. 

In metabolic diseases such as T2D and IR, IL1RAP has been described to be involved in the macrophage-mediated chronic inflammatory response [33]. In AT, the inflammatory response is mostly mediated by infiltrated macrophages that play a relevant role in controlling its functionality [42] and thus we consider that, in human AT, IL1RAP would play a relevant role in macrophage-mediated inflammatory process regulation.

In the present study, even if both miR221 and miR-222 are known to usually perform their regulation jointly, as they are part of the same cluster, we analyzed their expression levels in relation to obesity, diabetes, sex, and fat depot, considering at the same time each of the four variables, obesity, T2D, sex, and AT depot. Thus, the multivariate analysis highlighted that 20% and 18.8% of the expression levels of each miRNA in humans, respectively, and 30.8% and 27.4% of the expression levels in mice, respectively, were affected positively by diabetes and negatively by the variance among different tissue types. These data confirmed that the involvement of the miR221/miR-222 cluster in AT regulation should be studied considering the interaction of the two miRNAs (miR221 and miR-222) within tissues. In this regard, Kabekkodu [43] reported that members of an miRNA group presented different expression levels and co-expression patterns, indicating the possible existence of miRNA–miRNA interactions between groups and intra-groups, and that the expression of each miRNA within the same group depends on the expression of other cluster members. 

Moreover, we describe herein that the human AT *DVL2* expression level is fat depot and obesity dependent, which is in line with previous data relating this gene to insulin sensibility [44], and we provide new data pointing to the fact that *ETS1* and *IL1RAP* expression levels seem to be also related to fat depot. Collectively, our data suggest that the miR-221-3p/222-3p cluster could potentially regulate AT functionality through the modulation of *DVL2*, *ETS1*, and *IL1RAP* during obesity and T2D.

An innovative aspect of our study lies in the application of multivariate analysis to investigate the interactions between VAT and SAT, the miR-221-3p/222-3p cluster, and the identification of common potential target genes. This enabled us to pinpoint 3 significant common target genes (*ETS1*, *DVL2*, and *IL1RAP*) exhibiting markedly distinct expression profiles in both VAT and SAT among patients with obesity and T2D. Additionally, the binding site prediction analysis further supported the direct regulation of miR-221-3p and miR-222-3p in the gene regulatory networks of *ETS1*, *DVL2*, and *IL1RAP* concerning obesity and T2D.

It is important to highlight that the mouse model used in this research is well known for closely reflecting the human condition, particularly through shared dietary factors. Despite the significant clinical relevance of the human study, the in vivo investigations conducted in this animal model play a crucial role, enabling controlled experimentation and a thorough exploration of the underlying biological processes. We believe that the in vivo studies presented in this work serve to lay the groundwork for future extensive research on the molecular mechanisms governing the involvement of the miRNA cluster and its network in regulating AT functionality, particularly in the context of obesity and diabetes.

## 4. Materials and Methods

### 4.1. Patients, Study Design, and AT Collection

The minimum total sample size was determined as 33 patients and calculated using Epidat 4.2; 11 patients were assigned to each group. This sample size was sufficient to obtain a power of 90%, confidence level of 95%, and detect a difference of means = 0.015 and mean standard deviation = 0.01 (data obtained from our previous work [18]). 

The study included obese patients with BMI = 30–40 Kg/m^2^ and normal weight subjects (BMI = 20–25 Kg/m^2^) who underwent laparoscopic surgery for hiatus hernia or cholelithiasis at the Virgen de la Victoria University Hospital (Málaga, Spain). 

The criteria to assign participants to the different groups were as follow: normoglycemic (NG) with blood glucose levels after at least 8 h of fasting ≤ 110 mg/dL and with homeostatic model assessment for IR (HOMA-IR) < 4, and T2D with glycemia > 110 mg/dL and HOMA-IR > 7. Then, participants were classified into 3 groups according to their BMI, HOMA-IR, and glycemic status as NG-normoweight subjects (NG-NW), NG-obese subjects (NG-OB), and T2D-obese subjects (D-OB). In the D-OB group, 50% of subjects with and without Metformin treatment were included. The exclusion criteria were patients who had insulin treatment for T2D, prior cardiovascular disease, acute or chronic inflammatory disease, or infectious disease, patients who refused to participate in the study, and patients who had HOMA-IR between 4 and 7. 

Both SAT and VAT tissue samples were obtained during surgical procedures from abdominal and omental regions, respectively. Biopsy samples were washed in physiological saline solution, promptly frozen in liquid nitrogen, and stored at −80 °C until the assays were performed. All participants provided written informed consent and the study was reviewed and approved by the Ethics and Research Committee of the Virgen de la Victoria University Hospital (Málaga, Spain).

### 4.2. Laboratory Analysis

After an overnight fast and before surgery, blood samples were obtained from human subjects from the antecubital vein and placed in vacutainer tubes (VACUETTE TUBE Ref. 455045. Greiner Bio-One GmbH. Bad Haller Str. 32 4550, Kremsmünster, Austria). The serum was separated by centrifugation for 10 min at 4000 rpm and immediately frozen at −80 °C until analysis. Serum glucose, cholesterol, triglyceride, and HDL-cholesterol levels were measured using a Dimension auto-analyzer (Dade Behring Inc., Deerfield, IL, USA) with enzymatic methods (Randox Laboratories Ltd., Ardmore, 55 Diamond Road, County Antrim, UK). LDL-cholesterol level was calculated using the Friedewald equation. Insulin level was quantified by radioimmunoassay (BioSource International, 542 Flynn Rd, Camarillo, CA, USA) and IR was calculated using HOMA-IR, as previously described [45].

### 4.3. Generation of Diet-Induced Obese and Diabetic Mice

The C5776J strain mice used in the present study were purchased from Charles River, France (11 weeks old on arrival) and were allowed to acclimatize in the animal house for 1 week prior to the experiments. The mice were then housed individually under a 12 h light/dark cycle (8:00 p.m. lights off) in a temperature (21 ± 2 °C)- and humidity (50 ± 10%)-controlled room with free access to pelleted feed and water. The study was carried out with 3 groups of mice. The control mice were fed a standard diet (Standard Rodent Diet A04, SAFE, Panlab, Barcelona, Spain). The non-diabetic obese mice (45% HFD-ob) were fed a diet containing 10% Kcal (D12450 Research Diets Inc., New Brunswick, NJ, USA) for 6 weeks and then fed a high-density diet (D12451 Research Diets Inc., New Brunswick, NJ, USA) for another 8 weeks. The diabetic obese mice (45% HFD-D) were fed a high-density diet for 14 weeks (D12451 Research Diets Inc., New Brunswick, NJ, USA). Body weight was monitored twice weekly. After 14 weeks of feeding, glucose and insulin tolerance was assessed by intraperitoneal glucose tolerance tests (GTT and ITT). The mice were then euthanized by cervical dislocation. Inguinal SAT and VAT were immediately collected, frozen in liquid nitrogen, and stored at −80 °C for further analysis. 

### 4.4. MiRNA Extraction and Real-Time Quantitative PCR (qPCR)

MiRNAs from AT were isolated as previously described [18] using the mirVana™ miRNA Isolation Kit (Ambion life technologies, Carlsbad, CA, USA), according to the manufacturer’s guidelines. MiRNA concentration and purity were determined using a NanoDrop1000 spectrophotometer (Thermo Fischer Scientific, Inc., Franklin, MA, USA). cDNA was obtained using the TaqMan^®^ MicroRNA Reverse Transcription Kit (Applied Biosystems, Foster City, CA, USA) and specific primers and probes for each miRNA were used with the TaqMan^®^ MicroRNA Assay (Applied Biosystems): hsa-miR-221-3p (assay ID 000524) and hsa-miR-222-3p (assay ID 002276). Hsa-miR-16 (assay ID 000391) and snoRNA142 (assay ID 001231) were assessed using the Bestkeeper software to determine their usability as reference genes (http://www.wzw.tum.de/gene-quantification/bestkeeper.html; accessed on 18 May 2023) and used as the endogenous controls. A constant amount of 5 ng of miRNA was used to perform reverse transcription in a mixture containing 5 μL of RNA, 7 μL of RT-MasterMix, and 3 μL of RT-primers. The reverse transcription program consisted of 30 min at 16 °C, 30 min at 42 °C, and 5 min at 85 °C. miRNA expression levels were assessed by real-time qPCR using the Applied Biosystems 7500 Fast Real-Time PCR System (Applied Biosystems, Foster City, CA, USA). Each sample was assessed in duplicate and relative quantification of miRNA levels was performed using the comparative threshold cycle (Ct) method according to the manufacturer’s guidelines.

### 4.5. mRNA Isolation and qPCR 

The RNeasy^®^ Lipid Tissue Mini Kit (Qiagen, Washington, MD, USA) was used to isolate total mRNA from AT [18]. RNA concentration and purity were determined using a NanoDrop1000 spectrophotometer (Thermo Fischer Scientific, Inc.). cDNA synthesis was performed using the Transcriptor Reverse Transcriptase Kit (Roche Diagnostic, Barcelona, Spain) according to the manufacturer’s instructions, and real-time qPCR was performed using the Brillant III Ultra-Fast QPCR Master Mix (Agilent Technologies Ref.: 600880) with the Stratagene MX3005P QPCR System and V 4.1 software (Agilent Technologies). 

The reference gene (cyclophilin A) was selected using Bestkeeper software (http://www.gene-quantification.de/bestkeeper.html; accessed on 18 May 2023). The probes used for mRNA detection are detailed in Appendix A. Each sample was assessed in duplicate and relative quantification of mRNA levels was performed using the formula 2^−ΔCt^ according to the manufacturer’s guidelines.

### 4.6. Bioinformatics Analysis

The miRTarBase 4.0 web site (http://mirtarbase.mbc.nctu.edu.tw/; accessed on 16 July 2023) was used for identifying previously validated TGs of miR-221-3p and miR-222-3p. Moreover, the miRWalk 2.0 (http://www.umm.uni-heidelberg.de/apps/zmf/mirwalk/index.html; accessed on 16 July 2023) database was used for identifying predicted miR-221-3p and miR-222-3p TGs involved in angiogenesis, adipogenesis and apoptosis pathways. The PANTHER Database (Protein ANalysis Through Evolutionary Relationship) Classification System (http://www.pantherdb.org/; accessed on 17 July 2023) was applied to annotate the signaling pathways of validated target genes. GeneCodis3 (http://genecodis.dacya.ucm.es; accessed on 17 July 2023) was applied in the enrichment analysis of validated and non-validated target genes.

The interactions among miRNAs, biologic processes, and differentially expressed TGs were visualized using Cytoscape v.3.2.1 software (http://www.cytoscape.org/; accessed on 19 July 2023). 

The binding sites of miR-221-3p and miR-222-3p within the selected TGs were predicted using TargetScan Human 7.2, which indicated the conservation force (according to the position of the seed region), evolutionary conservation, and binding efficiency. 

### 4.7. Statistical Analysis

The results are expressed as the mean ± SEM. We tested the normality of the distribution of continuous variables using Shapiro–Wilk statistics. Data were analyzed using the Mann–Whitney U-test and Kruskal–Wallis or ANOVA and Student’s *t*-test for non-parametric or parametric data, respectively. Multivariate general linear and regression models were used in data analysis as follows: (a) obesity, T2D, type of AT depot, and different patient groups were introduced as independent variables; (b) expression levels of miR-221-3p and miR-222-3p or TGs were introduced as dependent variables. Associations between age and miRNA expression levels were analyzed using Pearson’s correlation. Statistical analyses were carried out using the statistical software package SPSS (version 22.0; SPSS Inc., Chicago, IL, USA). In order to decrease the probability of false positive results, we applied a *p*-value correction to obtain a stricter significance value. Therefore, a *p*-value ≤ 0.025 was considered statistically significant.

## 5. Conclusions

In conclusion, our findings underscore a prospective trajectory for future investigations, elucidating the intricate interplay between obesity, T2D, the miR-221-3p/222-3p cluster, and their target genes (ETS1, DVL2, and IL1RAP) in the modulation of AT functionality in the context of both T2D and obesity. These results have the potential to guide future research, facilitating the development of new treatment approaches designed for the management of both conditions.

## 6. Study Limitation

The limitation of this study lies in the substantially low sample size, which restricted the inclusion of too many covariates for result adjustment. The number of subjects allowed for the inclusion of age and HDLc as covariates; however, it was not feasible to include additional parameters such as non-diabetes-related medications.

## Figures and Tables

**Figure 1 ijms-24-17449-f001:**
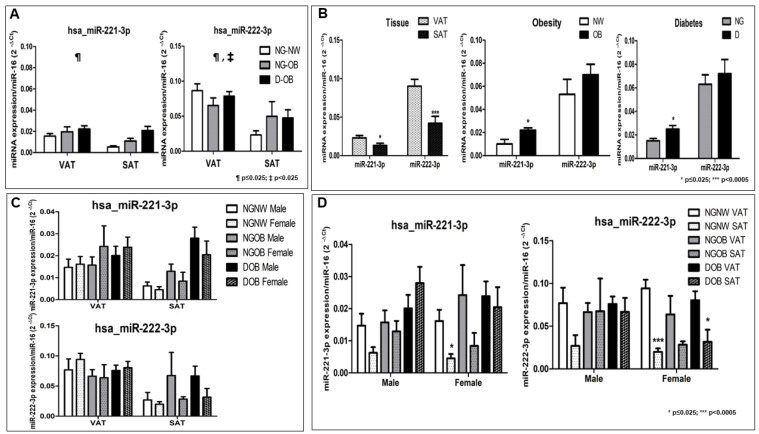
Expression profiles of miR-221-3p/222-3p cluster in human VAT and SAT. MiRNAs were extracted from VAT and SAT and their gene expression levels were measured via qPCR. Relative quantification of expression levels was performed using the comparative threshold cycle (Ct) method, with Hsa-miR-16 serving as the endogenous control. Data (*n* = 33) are presented as the mean ± SEM. (**A**,**B**) The data underwent analysis via a multivariate general linear model, incorporating tissue groups (VAT vs. SAT), diabetes status (NG vs. type 2 diabetic subjects), obesity status (NW vs. OB), and sex (male vs. female) as independent variables, with the expression levels of miR-221-3p and miR-222-3p as dependent variables and HDLc and age as co-variates. ¶ *p* ≤ 0.025 indicates a significant difference in the intercept of the model; ‡ *p* < 0.025 denotes a significant difference in tissue group (**A**). Pairwise comparisons were used to show the effect of tissue, obesity, and diabetes (**B**). (**C**,**D**) Expression profiles of miR-221-3p and 222-3p in human VAT and SAT were stratified by sex, and their expression levels were compared within the same tissue (**C**) and between both tissues, VAT vs. SAT (**D**). The Shapiro–Wilk test was employed to assess sample normality, and Student’s *t*-test was used for mean comparison. * *p* < 0.025; and *** *p* < 0.0005.

**Figure 2 ijms-24-17449-f002:**
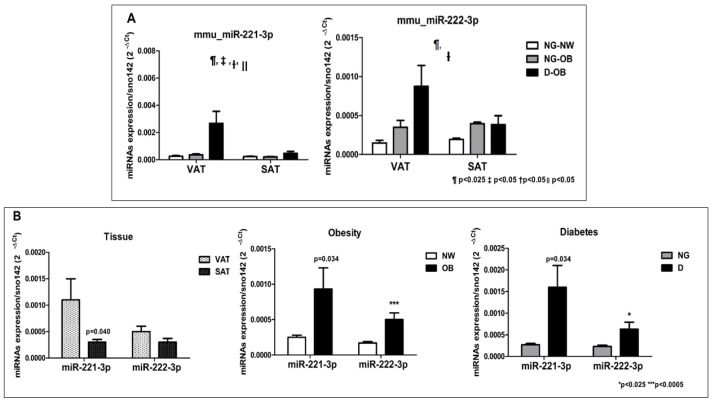
Expression profiles of miR-221-3p/222-3p cluster in mouse VAT and SAT. MiRNAs were isolated from VAT and SAT and their gene expression levels were assessed via qPCR. Relative quantification of the expression levels was conducted using the comparative threshold cycle (Ct) method, with SnoRNA142 serving as the endogenous control. Data (*n* = 18) are presented as the mean ± SEM. The data underwent analysis via a multivariate general linear model, incorporating tissue groups (VAT vs. SAT), diabetes status (NG vs. type 2 diabetic mice), and obesity status (NW vs. OB) as independent variables, with the expression levels of miR-221-3p and miR-222-3p as dependent variables. ¶ *p* < 0.025 shows a significant difference in the intercept of the model; ‡ *p* < 0.05 indicates a significant difference in tissue group; † *p* < 0.05 signifies a significant difference in the diabetic group; and ║ *p* < 0.05 denotes an interaction between diabetes and tissue (**A**). Pairwise comparisons were used to show the effect of tissue, obesity, and diabetes (**B**). * *p* < 0.025 and *** *p* < 0.0005.

**Figure 3 ijms-24-17449-f003:**
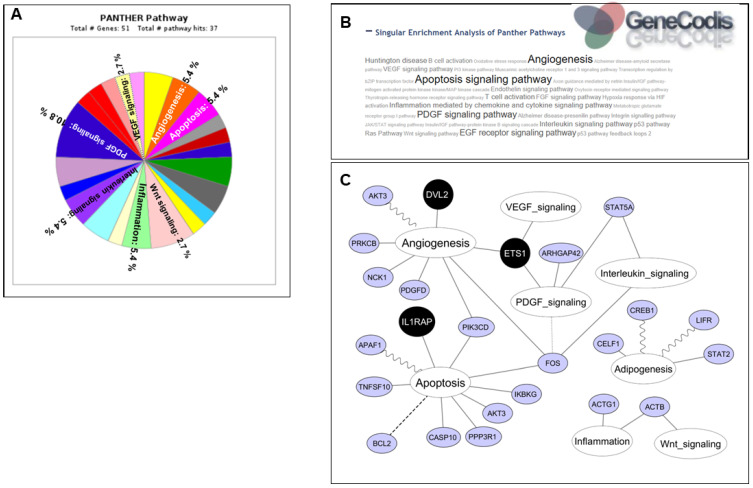
Bioinformatics analysis. (**A**) A set of 53 validated TGs, which are common targets for both miR-221-3p and miR-222-3p, were subjected to PANTHER analysis to elucidate their associated signaling pathways. (**B**) A compilation of 24 TGs, comprising 7 validated and 17 non-validated targets, involved in adipogenesis, angiogenesis, and apoptosis pathways were incorporated into GeneCodis3 for enrichment analysis. This analysis highlighted 3 TGs specifically implicated in angiogenesis and apoptosis pathways. (**C**) An interaction network was constructed using Cytoscape software. Pathways are represented by white nodes, while TGs are denoted by grey or black nodes. Edges delineate the involvement of miRNAs in the interaction between TGs and their corresponding biological functions, with solid lines indicating miR-221-3p and miR-222-3p, sine wave lines representing miR-221-3p, and dashed lines signifying miR-222-3p.

**Figure 4 ijms-24-17449-f004:**
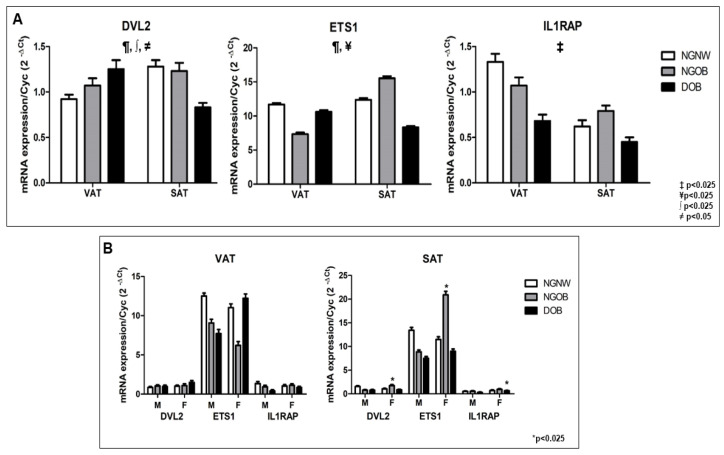
(**A**) mRNA expression levels of *ETS1*, *DVL2*, and *IL1RAP* in human VAT and SAT. Gene expression levels were assessed in VAT and SAT samples from NG-NW, NG-OB, and D-OB groups. Data are presented as the mean ± SEM. A multivariate general model was employed for statistical analysis. ¶ *p* < 0.025 shows a significant difference in the intercept of the model; ‡ *p* < 0.025 denotes a significant difference in tissue group; ¥ *p* < 0.025 signifies an interaction between obesity and sex; ∫ *p* < 0.025 indicates an interaction among obesity, sex, and tissue; and ≠ *p* < 0.05 denotes an interaction among tissue, diabetes, and sex. (**B**) Expression profiles of *DVL2*, *ETS1*, and *IL1RAP* in human VAT and SAT stratified by sex. Expression levels were compared between male and female samples for each tissue type. The Shapiro–Wilk test was employed to assess sample normality, and Student’s *t*-test was used for mean comparison. * *p* < 0.025.

**Table 1 ijms-24-17449-t001:** Anthropometric and biochemical characteristics of the study groups.

	NG-NW (*n* = 11)	NG-OB (*n* = 11)	D-OB (*n* = 11)
Age (years)	41.32 ± 4.11	46.91 ± 3.55	43.91 ± 4.13
Sex (Man/Woman)	5/6	5/6	5/6
BMI (kg/m^2^)	22.48 ± 0.46	37.48 ± 1.39 *	47.07 ± 2.52 *^#^
Glucose (mg/dL)	95.73 ± 4.05	89.91 ± 3.11	172.86 ± 22.73 *^#^
HOMA-IR	1.18 ± 1.56	2.28 ± 0.3 *	7.90 ± 0.75 *^#^
Triglycerides (mg/dL)	72.23 ± 6.29	144.45 ± 27.39 *	166.86 ± 20.45 *^#^
Cholesterol (mg/dL)	183.77 ± 5.15	202.05 ± 11.90	186.41 ± 14.99
HDL-c (mg/dL)	62.5 ± 3.83	47.82 ± 5.39 *	41.91 ± 2.65 *^#^
LDL-c (mg/dL)	111.36 ± 5.53	126.7 ± 9.54	115 ± 14.56

Study patients (*n* = 33) were selected according to BMI, HOMA-IR, and glycemic state. Data are expressed as the mean ± SEM. Comparison among groups was performed using a Mann–Whitney U test. Different superscript signs represent statistically significant differences between groups (*p* < 0.05), *: NG-OB or D-OB versus NG-NW; #: D-OB versus NG-OB. BMI: body mass index; HOMA index: homeostasis model assessment index; NG-NW: normoglycemic normoweight subjects; NG-OB: normoglycemic subjects with obesity; D-OB: diabetic subjects with obesity.

**Table 2 ijms-24-17449-t002:** Bioinformatics analysis.

	Validated Genes	Non-Validated Genes	Total
miRNA	miRTarBase	miRWalk	
miR-221	467	16	483
miR-222	444	13	457
TOTAL	911	29	940
COMMON	53	14	67
ENRICHMENT ANALYSIS	7	17	24
SELECTED	1	2	3

## Data Availability

All data generated or analyzed during this study are included in this article. Further enquiries can be directed to the corresponding author.

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
