# Peer review of "MiR-221-3p/222-3p Cluster Expression in Human Adipose Tissue Is Related to Obesity and Type 2 Diabetes"

_ijms, 2023, doi:10.3390/ijms242417449_

Round 1
Reviewer 1 Report
Comments and Suggestions for Authors
Please refer to the attached document.

Author Response
We would like to express our gratitude to the reviewer for providing us with detailed and constructive feedback. We have carefully considered their comments and believe that by addressing their points, we have made significant improvements to the article.
- For fig. 1A and related description, the p values or statistically significant values are nowhere to be seen on the graphs. Could the authors please include them on the graphs?
In the Figures 1A, 1B, and 1D the p-values or the statistically significant values are indicated by the symbols. Figure 1C does not feature symbols as there were no statistically significant differences in the analysis, as explained in the results section. Also, we have included the significant values on the graphs, as suggested by the reviewer.
- In line 127, by saying "negative effect of tissue type", do the authors mean that the SQAT or SAT has lower expression levels of these miRNAs?
Indeed, as the reviewer correctly interpreted, the expression level of both miRNAs is lower when obesity and diabetes are present, and this effect varies by tissue type. This is elucidated in the regression analysis by the negative Beta value, as explained in the Results section 2.2 and illustrated in Fig. 1B. In our specific case, the expression level of both miRNAs is lower in SAT when compared to VAT, while obesity and diabetes are present
- In results section 2.2, could the authors please explain how did they obtain these numbers i.e. 19.5% and 20%?
In a regression analysis, the R2 value indicates the proportion of variability in the dependent variable explained by the model. For example, if R2=0.195 or R2=0.20, it means that 19.5% or 20% of the variability in the dependent variable (miR-221 or miR-222) is explained by the independent variables in the model (Obesity, diabetes, tissue, etc)
- For fig. 2, the negative effect of the tissue is not very clear. Also, please elaborate how the numbers "36.9%" and "33.8%" were obtained.
We apologize if this issue was not clear, and we appreciate the reviewer's input. For this reason, we have now included Figure 2B, where both the negative effect of tissue and the positive effects of obesity and diabetes can be clearly observed. The values of 36.9% and 33.8% are derived from the R2 value in the regression analysis, as explained in the response 3.
- For fig. 4, if the levels of these two miRNA's were elevated in DOB conditions as shown in Fig. 1 A, why did the pattern not repeat itself during the analysis of their target genes? Is it because the trends seen in Fig. 1A never reaches statistical significance?
The reviewer makes an interesting observation; however, it's not necessary for the expression patterns of miRNAs and the genes they regulate to be identical. As is well known, a single miRNA can regulate the expression of multiple genes, either increasing or decreasing their expression through canonical or non-canonical pathways. Additionally, a specific gene can be regulated by several miRNAs. This complexity can lead to gene and miRNA expression patterns that are not identical. MiRNAs and their regulated genes may have different temporal responses. MiRNA expression can change rapidly in response to cellular signals, whereas gene expression may be slower and more prolonged. Therefore, miRNAs may exhibit expression patterns opposite to the genes they regulate. This review could serve as an example to illustrate the different patterns that miRNAs and their respective target genes and pathways can exhibit: https://pubmed.ncbi.nlm.nih.gov/29291020/
- Also, did the authors perform a combined analysis of VAT and SAT for each sex as shown in Fig. 1 C and D?
If we understand the reviewer correctly regarding a combined analysis of VAT and SAT for each sex, considering the known variations in fat distribution and the influence of sex hormones, we hypothesized that the functions of these miRNAs in subcutaneous adipose tissue (SAT) and visceral adipose tissue (VAT) might differ based on sex. Hence, Figure 1C compares the expression level of the miRNA in each sex within the same tissue (VAT or SAT), while Figure 1D compares the expression level of the miRNA for each sex in different tissues (VAT vs. SAT).
If the reviewer is referring to comparing the expression level of miRNA in VAT+SAT by sex, that analysis corresponds to the multivariate analysis shown in Figure 1A, where the sex variable was used as an independent variable, and there was no significant difference in the expression of both miRNAs related to sex when VAT and SAT are present simultaneously.
- Also, Are these the same patients that were studied to generate data from Fig. 1? If not, could the authors please explain how many patients were included in this data?
In fact, all the data have been obtained from the same group of patients.
- In methods section 4.3, could the authors please mention which adipose depots were collected?
We apologize if this was not adequately mentioned in the text. Adipose tissues collected included inguinal adipose tissue (SAT) and visceral adipose tissue. This information is now explicitly stated in the Methods section. (line 440)
- What do the authors want to convey by mentioning the symbols in fig. 1A , fig. 2 as well as 4 A? Should they be on each miRNA/gene depending on the contributing factor of variance? It is not clear if it applies to all the genes/both miRNAs or just one.
We apologize if the presentation of data in these figures could potentially lead to misinterpretation or misunderstanding of the results. In response to the reviewer's inquiry about the interpretation of symbols and their placement, we have proactively adjusted their positioning above each miRNA (Fig. 1A and 2A), considering the specific factor of variance, as accurately highlighted by the reviewer. Additionally, Figure 4A has been revised to incorporate symbols above each gene, indicating the corresponding contributing factor of variance.
- Please include a rationale for mouse studies when more relevant human study was already undertaken.
We appreciate the reviewer's concern and recognize the significance of human studies. While the human study provides crucial clinical relevance, mouse studies contribute by allowing controlled experimentation and a more in-depth exploration of underlying biological processes. In this context, our objective was to investigate whether the network of miRNA clusters, along with their respective pathways and target genes, follows a similar pattern in the mouse model as observed in humans. This exploration aims to pave the way for future comprehensive studies on the molecular mechanisms governing the involvement of this miRNA cluster and its network in regulating adipose tissue functionality, particularly in the context of obesity and diabetes. It's important to emphasize that this mouse model of obesity and diabetes closely mirrors the human scenario, as both conditions are associated with dietary factors.
Furthermore, it's worth noting that in humans, studies on the underlying molecular mechanisms are challenging to conduct, except in vitro. In contrast, in mice, in vivo studies can be performed, typically using inhibitors or mimics of this miRNA cluster. This approach can provide reliable data, offering valuable insights for the exploration of potential therapeutic interventions.
We would like to express our gratitude to the reviewer for providing us with detailed and constructive feedback. We have carefully considered their comments and believe that by addressing their points, we have made significant improvements to the article.
- For fig. 1A and related description, the p values or statistically significant values are nowhere to be seen on the graphs. Could the authors please include them on the graphs?
In the Figures 1A, 1B, and 1D the p-values or the statistically significant values are indicated by the symbols. Figure 1C does not feature symbols as there were no statistically significant differences in the analysis, as explained in the results section. Also, we have included the significant values on the graphs, as suggested by the reviewer.
- In line 127, by saying "negative effect of tissue type", do the authors mean that the SQAT or SAT has lower expression levels of these miRNAs?
Indeed, as the reviewer correctly interpreted, the expression level of both miRNAs is lower when obesity and diabetes are present, and this effect varies by tissue type. This is elucidated in the regression analysis by the negative Beta value, as explained in the Results section 2.2 and illustrated in Fig. 1B. In our specific case, the expression level of both miRNAs is lower in SAT when compared to VAT, while obesity and diabetes are present
- In results section 2.2, could the authors please explain how did they obtain these numbers i.e. 19.5% and 20%?
In a regression analysis, the R2 value indicates the proportion of variability in the dependent variable explained by the model. For example, if R2=0.195 or R2=0.20, it means that 19.5% or 20% of the variability in the dependent variable (miR-221 or miR-222) is explained by the independent variables in the model (Obesity, diabetes, tissue, etc)
- For fig. 2, the negative effect of the tissue is not very clear. Also, please elaborate how the numbers "36.9%" and "33.8%" were obtained.
We apologize if this issue was not clear, and we appreciate the reviewer's input. For this reason, we have now included Figure 2B, where both the negative effect of tissue and the positive effects of obesity and diabetes can be clearly observed. The values of 36.9% and 33.8% are derived from the R2 value in the regression analysis, as explained in the response 3.
- For fig. 4, if the levels of these two miRNA's were elevated in DOB conditions as shown in Fig. 1 A, why did the pattern not repeat itself during the analysis of their target genes? Is it because the trends seen in Fig. 1A never reaches statistical significance?
The reviewer makes an interesting observation; however, it's not necessary for the expression patterns of miRNAs and the genes they regulate to be identical. As is well known, a single miRNA can regulate the expression of multiple genes, either increasing or decreasing their expression through canonical or non-canonical pathways. Additionally, a specific gene can be regulated by several miRNAs. This complexity can lead to gene and miRNA expression patterns that are not identical. MiRNAs and their regulated genes may have different temporal responses. MiRNA expression can change rapidly in response to cellular signals, whereas gene expression may be slower and more prolonged. Therefore, miRNAs may exhibit expression patterns opposite to the genes they regulate. This review could serve as an example to illustrate the different patterns that miRNAs and their respective target genes and pathways can exhibit: https://pubmed.ncbi.nlm.nih.gov/29291020/
- Also, did the authors perform a combined analysis of VAT and SAT for each sex as shown in Fig. 1 C and D?
If we understand the reviewer correctly regarding a combined analysis of VAT and SAT for each sex, considering the known variations in fat distribution and the influence of sex hormones, we hypothesized that the functions of these miRNAs in subcutaneous adipose tissue (SAT) and visceral adipose tissue (VAT) might differ based on sex. Hence, Figure 1C compares the expression level of the miRNA in each sex within the same tissue (VAT or SAT), while Figure 1D compares the expression level of the miRNA for each sex in different tissues (VAT vs. SAT).
If the reviewer is referring to comparing the expression level of miRNA in VAT+SAT by sex, that analysis corresponds to the multivariate analysis shown in Figure 1A, where the sex variable was used as an independent variable, and there was no significant difference in the expression of both miRNAs related to sex when VAT and SAT are present simultaneously.
- Also, Are these the same patients that were studied to generate data from Fig. 1? If not, could the authors please explain how many patients were included in this data?
In fact, all the data have been obtained from the same group of patients.
- In methods section 4.3, could the authors please mention which adipose depots were collected?
We apologize if this was not adequately mentioned in the text. Adipose tissues collected included inguinal adipose tissue (SAT) and visceral adipose tissue. This information is now explicitly stated in the Methods section. (line 440)
- What do the authors want to convey by mentioning the symbols in fig. 1A , fig. 2 as well as 4 A? Should they be on each miRNA/gene depending on the contributing factor of variance? It is not clear if it applies to all the genes/both miRNAs or just one.
We apologize if the presentation of data in these figures could potentially lead to misinterpretation or misunderstanding of the results. In response to the reviewer's inquiry about the interpretation of symbols and their placement, we have proactively adjusted their positioning above each miRNA (Fig. 1A and 2A), considering the specific factor of variance, as accurately highlighted by the reviewer. Additionally, Figure 4A has been revised to incorporate symbols above each gene, indicating the corresponding contributing factor of variance.
- Please include a rationale for mouse studies when more relevant human study was already undertaken.
We appreciate the reviewer's concern and recognize the significance of human studies. While the human study provides crucial clinical relevance, mouse studies contribute by allowing controlled experimentation and a more in-depth exploration of underlying biological processes. In this context, our objective was to investigate whether the network of miRNA clusters, along with their respective pathways and target genes, follows a similar pattern in the mouse model as observed in humans. This exploration aims to pave the way for future comprehensive studies on the molecular mechanisms governing the involvement of this miRNA cluster and its network in regulating adipose tissue functionality, particularly in the context of obesity and diabetes. It's important to emphasize that this mouse model of obesity and diabetes closely mirrors the human scenario, as both conditions are associated with dietary factors.
Furthermore, it's worth noting that in humans, studies on the underlying molecular mechanisms are challenging to conduct, except in vitro. In contrast, in mice, in vivo studies can be performed, typically using inhibitors or mimics of this miRNA cluster. This approach can provide reliable data, offering valuable insights for the exploration of potential therapeutic interventions.

Reviewer 2 Report
Comments and Suggestions for Authors
The study by Gentile and colleagues aims to investigate the relationship between the miR-221-3p/222-3p cluster in human AT and its regulatory network with obesity and T2D. Expression levels of miR-221-3p/222-3p and their target genes (TG) were measured by qPCR in visceral (VAT) and subcutaneous (SAT) ATs from patients (n=33) with normal body weight and obesity based on BMI, and normoglycemic (NG) and type 2 diabetes (T2D) based on glycemic status. Their results suggest that the miR-221-3p/222-3p cluster and its associated regulatory networks may be a viable target for the treatment of obesity and related metabolic disorders.
Comments and suggestions:
- Several abbreviations were not introduced before their use. Please review the manuscript carefully.
- In Table 1, decimal places are marked with a comma, but in English, they should be marked with a dot.
- Insulin levels are not shown in Table 1 and LDL levels are incorrect for the NG-NW group.
- Why were three and not four groups formed? The diabetic but not obesity subgroup is missing.
- In many statistical analyses, age and gender are missing as covariates. Since there are no differences between subgroups in these variables, I recommend their use in the analyses.
- What was the reference for tissue type?
- As the results show the simultaneous testing of several genetic elements, I suggest using a test correction for the p-value (0.05/2 = 0.025).
- In the linear regression analyses, did all the parameters tested show a normal distribution?
- Did the study group consider the use of medicines unrelated to diabetes (hypertension, lipid-lowering drugs, etc.)? Many drugs have the potential to affect lipid metabolism, thus biasing the results of the present study.
- For the diathetic group, those receiving insulin treatment were excluded. However, many (non-insulin) treatments are used for T2D patients. Were these considered in the selection and/or analyses?
- HDL cholesterol levels are strongly associated with T2D and showed a difference between the two groups in the present study. I suggest its use as a covariate in the analyses.
Author Response
The study by Gentile and colleagues aims to investigate the relationship between the miR-221-3p/222-3p cluster in human AT and its regulatory network with obesity and T2D. Expression levels of miR-221-3p/222-3p and their target genes (TG) were measured by qPCR in visceral (VAT) and subcutaneous (SAT) ATs from patients (n=33) with normal body weight and obesity based on BMI, and normoglycemic (NG) and type 2 diabetes (T2D) based on glycemic status. Their results suggest that the miR-221-3p/222-3p cluster and its associated regulatory networks may be a viable target for the treatment of obesity and related metabolic disorders.
We would like to express our gratitude to the reviewer for providing us with detailed and constructive feedback. We have carefully considered their comments and believe that by addressing their points, we have made significant improvements to the article.
Comments and suggestions:
- Several abbreviations were not introduced before their use. Please review the manuscript carefully.
We apologize for the oversight in not introducing several abbreviations before their use. We have now addressed it
- In Table 1, decimal places are marked with a comma, but in English, they should be marked with a dot.
We apologize for the mistake in Table 1 where decimal places were marked with a comma instead of a dot, as is appropriate in English. We have now corrected this and replaced the comma with a dot.
- Insulin levels are not shown in Table 1 and LDL levels are incorrect for the NG-NW group.
Indeed, we did not display insulin levels in the table as we deemed HOMA-IR to be a more indicative measure for patient stratification. Additionally, the LDL value has already been corrected in the table.
- Why were three and not four groups formed? The diabetic but not obesity subgroup is missing.
We find the reviewer's observation to be very interesting. In fact, including non-obese patients with diabetes in the study could provide a more comprehensive representation of the population. This would be especially necessary if the goal were to identify specific risk factors for diabetes beyond obesity. However, the study is specifically designed to investigate factors related to both obesity and diabetes. The inclusion of additional groups could demand more resources and increase the complexity of the study. Additionally, it would require additional time, as recruiting this patient group is challenging. As we mentioned before, the study is not designed to include this particular patient group.
- In many statistical analyses, age and gender are missing as covariates. Since there are no differences between subgroups in these variables, I recommend their use in the analyses.
We appreciate the reviewer's recommendation. We have now included age and gender as covariates in all analyses. This analysis has been explained in the text, and the results are presented (line 118)
- What was the reference for tissue type?
We did not fully understand the reviewer's question. If it pertains to understanding the reference for the tissue type, in our study, the tissue reference is the control group. We have adhered to this practice within our research group for several years. In the design of our study, we chose to include the same type of tissue (VAT/SAT) found in the control group (without obesity/without diabetes) for analysis. This tissue serves as a baseline reference for comparing gene expression changes in individuals with obesity and/or diabetes. It is crucial to note that the control group is carefully selected to match its characteristics with the obesity and diabetes groups included in the study. In summary, the study encompasses three types of visceral or subcutaneous adipose tissues (VAT or SAT): Normal VAT/SAT (without obesity and without diabetes) (reference); Obesity VAT/SAT (from obese patients without diabetes), and Diabetes VAT/SAT (from obese patients with diabetes). If our response has not adequately addressed your question, please clarify, and we will provide a more suitable answer."
- As the results show the simultaneous testing of several genetic elements, I suggest using a test correction for the p-value (0.05/2 = 0.025).
We appreciate the reviewer's suggestion. We have now employed a corrected p-value, as explained in the Materials and Methods
- In the linear regression analyses, did all the parameters tested show a normal distribution?
In fact, all the tested parameters exhibit a normal distribution. This information is presented in the supplementary figures S1A and S1B (histogram and Q-Q plot), with detailed results available in the Results section.
- Did the study group consider the use of medicines unrelated to diabetes (hypertension, lipid-lowering drugs, etc.)? Many drugs have the potential to affect lipid metabolism, thus biasing the results of the present study.
We greatly appreciate the reviewer's suggestion and reflection. However, we have noted that information on the use of medications unrelated to diabetes is not comprehensively recorded in our database, preventing us from incorporating this variable as an adjustment factor in our statistical analysis. Despite this limitation, we identify a strength in our study: the patient cohort is highly homogeneous. This homogeneity lends robustness to our results, mitigating the challenge of a relatively small sample size that limits the inclusion of more adjustment variables in the analysis. Regarding lipid metabolism, we examined factors such as triglycerides, total cholesterol, and LDLc in both patient groups (NG-OB and D-OB). We observed no significant differences in these factors between the two groups, reinforcing the cohort's homogeneity. However, a significant difference in HDLc levels between the NG-OB and D-OB groups was identified (line 119). Since this is the only factor showing a significant difference in lipid metabolism, we chose to use it as an adjustment variable in our statistical analysis. Despite including HDLc and age as control variables, we acknowledge that introducing more confounding factors could be beneficial. Nevertheless, due to the sample size limitation, we have presented this as a study limitation. This issue is mentioned in the text (lines 510-514).
In summary, despite these limitations, we believe our results are reliable due to the cohort's homogeneity. The inclusion of control variables provides a solid foundation for drawing conclusions from our study, offering valuable insights into the relationship between diabetes, obesity, and lipid metabolism in our study population.
- For the diathetic group, those receiving insulin treatment were excluded. However, many (non-insulin) treatments are used for T2D patients. Were these considered in the selection and/or analyses?
We appreciate the reviewer's inquiry. Patients in the D-OB group were recruited in equal proportions (50% with/50% without) with respect to metformin treatment. This condition has been specified in the Materials and Methods section (Lines 404-406)
- HDL cholesterol levels are strongly associated with T2D and showed a difference between the two groups in the present study. I suggest its use as a covariate in the analyses.
We agree with the reviewer and appreciate the suggestion. The HDL cholesterol variable, along with age, has been introduced as covariates in the statistical analysis to reduce redundant variability and enhance the precision of parameter estimates in the model. Most of the results have remained with significant differences or followed the trend with an almost significant outcome (with corrected p-values), demonstrating their reliability. As we have been explaining to the reviewer, we believe the strength of this study lies in a highly homogeneous cohort. We acknowledge that achieving statistical significance for these parameters would require a larger sample size. However, recruiting patients to form such a homogeneous group, as in this study, is challenging.

Reviewer 3 Report
Comments and Suggestions for Authors
I would like to thank Gentile et al for this interesting paper on the function of specific microRNAs in adipose tissue in obesity and type 2 diabetes. In total, this is a well conducted study and I have minor suggestions:
Why Materials and Methods section is located after Discussion? It should be replaced after the Introduction section.
The last sentence of the Abstract is not substantiated by the study's results ("these findings suggest...") and should be acccordingly rephrased.
Line 57: "was extensively proven to be involved..."
Line 62: expands
Line 63: "resistance and T2D"
Line 63: "is due to its"
Line 66: "which prevents hypoxia"
Line 95-97: please rephrase. Also explain the acronyms before using them
Line 97: "Additionally" is misplaced since the second sentence is not related to the first
Line 101: "Donors" is not mentioned before and should be better explained (probably because Materials and Methods was misplaced after Results)
Line 267: "in the presence of both male and female sex"
In Discussion there are some really long sentences (for ex. lines 283-287) that should be broken down to two sentences, since they are difficult to follow
Line 474: "highlight a promising avenue for future research": please rephrase to a more academic style
Comments on the Quality of English LanguageMinor revisions as suggested
Author Response
I would like to thank Gentile et al for this interesting paper on the function of specific microRNAs in adipose tissue in obesity and type 2 diabetes. In total, this is a well conducted study and I have minor suggestions:
We would like to express our gratitude to the reviewer for providing us with detailed and constructive feedback. We have carefully considered their comments and believe that by addressing their points, we have made significant improvements to the article.
- Why Materials and Methods section is located after Discussion? It should be replaced after the Introduction section.
We appreciate the reviewer's observation. We want to point out that we have followed the format required by the journal, which mandates placing the Materials and Methods section after discussion section.
2.The last sentence of the Abstract is not substantiated by the study's results ("these findings suggest...") and should be acccordingly rephrased.
The final sentence of the Abstract has been modified in response to the reviewer's suggestion (Lines 49-52)
3.Line 57: "was extensively proven to be involved..."
Ok.
Line 62: expands
ok
Line 63: "resistance and T2D"
ok
Line 63: "is due to its"
ok
Line 66: "which prevents hypoxia"
ok
Line 95-97: please rephrase. Also explain the acronyms before using them
We have reconsidered the highlighted paragraph as suggested by the reviewer, and we have also provided explanations for the acronyms before their use (Lines 99-100)
Line 97: "Additionally" is misplaced since the second sentence is not related to the first
The sentence was corrected.
Line 101: "Donors" is not mentioned before and should be better explained (probably because Materials and Methods was misplaced after Results)
“Donors" was removed and replaced with "Study patients.”
Line 267: "in the presence of both male and female sex"
We appreciate the reviewer for this observation. We have corrected this error.
In Discussion there are some really long sentences (for ex. lines 283-287) that should be broken down to two sentences, since they are difficult to follow
We apologize for having formulated lengthy sentences. Long sentences were shortened to enhance the clarity of the discussion.
Line 474: "highlight a promising avenue for future research": please rephrase to a more academic style
We appreciate the reviewer's suggestion. The formulation of this paragraph has been revised.

Round 2
Reviewer 1 Report
Comments and Suggestions for Authors
Please refer to the attached documents.

Author Response
We would like to express our gratitude to the reviewer for providing us with detailed and constructive feedback. We have carefully considered their comments and believe that by addressing their points, we have made significant improvements to the article.
- For fig. 1A and related description, the p values or statistically significant values are nowhere to be seen on the graphs. Could the authors please include them on the graphs?
In the Figures 1A, 1B, and 1D the p-values or the statistically significant values are indicated by the symbols. Figure 1C does not feature symbols as there were no statistically significant differences in the analysis, as explained in the results section. Also, we have included the significant values on the graphs, as suggested by the reviewer.
- In line 127, by saying "negative effect of tissue type", do the authors mean that the SQAT or SAT has lower expression levels of these miRNAs?
Indeed, as the reviewer correctly interpreted, the expression level of both miRNAs is lower when obesity and diabetes are present, and this effect varies by tissue type. This is elucidated in the regression analysis by the negative Beta value, as explained in the Results section 2.2 and illustrated in Fig. 1B. In our specific case, the expression level of both miRNAs is lower in SAT when compared to VAT, while obesity and diabetes are present.
The explanation does answer my question. However, the term “negative effect of
tissue type” indicates a causative effect of the tissue type per se on the expression levels. There could be other factors that exert their effect on the expression levels in different tissue types, and not the tissues itself regulating the expression levels. In other words, it is not clear whether the tissues themselves exert such effect and hence, my suggestion is to replace this term with something more generic such as “variance among different tissue types”.
***We apologize if this issue was not clear, and we appreciate the reviewer's input. To address this concern, we have revised the text to mention 'variance among different tissue types' instead of the previously stated 'negative effect of tissue types.' This modification has been made in the following lines: 134, 137, 196, 290, and 383."
- In results section 2.2, could the authors please explain how did they obtain these numbers i.e.
19.5% and 20%?
In a regression analysis, the R2 value indicates the proportion of variability in the dependent variable explained by the model. For example, if R2=0.195 or R2=0.20, it means that 19.5% or 20% of the variability in the dependent variable (miR-221 or miR-222) is explained by the independent variables in the model (Obesity, diabetes, tissue, etc)
Is there a way to graphically represent this data, if not in the main manuscript, then may be in the supplementary information? This applies to every mention of such information within the manuscript.
***We have attempted to present this information in graphical form, as suggested by the reviewer. This information is now illustrated in the supplementary figures: S1A, S1B, and S1C for hsa-miR-221/222; S2A and S2B (histogram and Q-Q plot); S3A, S3B, and S3C for mmu-miR-221/222; S4A and S4B (histogram and Q-Q plot). Detailed results can be found in the Results section: 2.2 and 2.4.
- For fig. 2, the negative effect of the tissue is not very clear. Also, please elaborate how the numbers "36.9%" and "33.8%" were obtained.
We apologize if this issue was not clear, and we appreciate the reviewer's input. For this reason, we have now included Figure 2B, where both the negative effect of tissue and the positive effects of obesity and diabetes can be clearly observed. The values of 36.9% and 33.8% are derived from the R2 value in the regression analysis, as explained in the response 3.
In Fig. 2B, for depicting the effect of obesity and diabetes, which tissue was
used? Was it SAT or VAT?
***We apologize if the matter was unclear, and we appreciate the reviewer's input. The reviewer has requested clarification regarding the specific impact of obesity and diabetes on a particular adipose tissue location. In response, we have investigated the effects of obesity and T2D on miRNA expression, conducting analyses base on adipose tissue location. Each miRNA was considered a dependent variable in separate models, with obesity and T2D serving as independent variables categorized by tissue type (Fig. S5A, B, C, and D). Detailed results can be found in the Results section.
- For fig. 4, if the levels of these two miRNA's were elevated in DOB conditions as shown in Fig. 1 A, why did the pattern not repeat itself during the analysis of their target genes? Is it because the trends seen in Fig. 1A never reaches statistical significance?
The reviewer makes an interesting observation; however, it's not necessary for the expressionpatterns of miRNAs and the genes they regulate to be identical. As is well known, a single miRNA can regulate the expression of multiple genes, either increasing or decreasing their expression through canonical or non-canonical pathways. Additionally, a specific gene can be regulated by several miRNAs. This complexity can lead to gene and miRNA expression patterns that are not identical. MiRNAs and their regulated genes may have different temporal responses. MiRNA expression can change rapidly in response to cellular signals, whereas gene expression may be slower and more prolonged. Therefore, miRNAs may exhibit expression patterns opposite to the genes they regulate. This review could serve as an example to illustrate the different patterns that miRNAs and their respective target genes and pathways can exhibit:https://pubmed.ncbi.nlm.nih.gov/29291020/ Could the authors please put this explanation in discussion or cite this review?
***We appreciate the reviewer's suggestion. This explanation has now been included into the text, and the corresponding reference has been added.
- Also, did the authors perform a combined analysis of VAT and SAT for each sex as shown in Fig. 1 C and D?
If we understand the reviewer correctly regarding a combined analysis of VAT and SAT for each sex, considering the known variations in fat distribution and the influence of sex hormones, we hypothesized that the functions of these miRNAs in subcutaneous adipose tissue (SAT) and visceral adipose tissue (VAT) might differ based on sex. Hence, Figure 1C compares the expression level of the miRNA in each sex within the same tissue (VAT or SAT), while Figure 1D compares the expression level of the miRNA for each sex in different tissues (VAT vs. SAT).
If the reviewer is referring to comparing the expression level of miRNA in VAT+SAT by sex, that analysis corresponds to the multivariate analysis shown in Figure 1A, where the sex variable was used as an independent variable, and there was no significant difference in the expression of both miRNAs related to sex when VAT and SAT are present simultaneously.
The question was whether panel A (both in the old and new version) in Fig. 4
was specific for any sex? Or was it combined from both males and females for
that particular tissue type?
***Indeed, as the reviewer correctly interpreted, Panel A in Fig. 4 was combined from both males and females for that specific tissue type.
- Also, Are these the same patients that were studied to generate data from Fig. 1? If not, could the authors please explain how many patients were included in this data?
In fact, all the data have been obtained from the same group of patients.
- In methods section 4.3, could the authors please mention which adipose depots were collected?
We apologize if this was not adequately mentioned in the text. Adipose tissues collected included inguinal adipose tissue (SAT) and visceral adipose tissue. This information is now explicitly stated in the Methods section. (line 440)
- What do the authors want to convey by mentioning the symbols in fig. 1A , fig. 2 as well as 4 A? Should they be on each miRNA/gene depending on the contributing factor of variance? It is not clear if it applies to all the genes/both miRNAs or just one.
We apologize if the presentation of data in these figures could potentially lead to
misinterpretation or misunderstanding of the results. In response to the reviewer's inquiry about the interpretation of symbols and their placement, we have proactively adjusted their positioning above each miRNA (Fig. 1A and 2A), considering the specific factor of variance, as accurately highlighted by the reviewer. Additionally, Figure 4A has been revised to incorporate symbols above each gene, indicating the corresponding contributing factor of variance.
- Please include a rationale for mouse studies when more relevant human study was already undertaken.
We appreciate the reviewer's concern and recognize the significance of human studies. While the human study provides crucial clinical relevance, mouse studies contribute by allowing controlled experimentation and a more in-depth exploration of underlying biological processes. This exploration aims to pave the way for future comprehensive studies on the molecular mechanisms governing the involvement of this miRNA cluster and its network in regulating adipose tissue functionality, particularly in the context of obesity and diabetes. It's important to emphasize that this mouse model of
obesity and diabetes closely mirrors the human scenario, as both conditions are associated with dietary factors. Furthermore, it's worth noting that in humans, studies on the underlying molecular mechanisms are challenging to conduct, except in vitro. In contrast, in mice, in vivo studies can be performed, typically using inhibitors or mimics of this miRNA cluster. This approach can provide reliable data, offering valuable insights for the exploration of potential therapeutic interventions.
Could the authors please put this explanation in discussion?
***We appreciate the reviewer's suggestion. We have now included it in the discussion.

Reviewer 2 Report
Comments and Suggestions for Authors
I accept the Authors' answers to my questions and comments.
Author Response
We greatly appreciate the reviewer's support in improving our manuscript and the constructive feedback provided.
Round 3
Reviewer 1 Report
Comments and Suggestions for Authors
The authors have satisfactorily addressed all of the previous concerns and comments.